# QCNN_BaOpt: Multi-Dimensional Data-Based Traffic-Volume Prediction in Cyber–Physical Systems

**DOI:** 10.3390/s23031485

**Published:** 2023-01-29

**Authors:** Ramesh Sneka Nandhini, Ramanathan Lakshmanan

**Affiliations:** School of Computer Science and Engineering, Vellore Institute of Technology, Vellore 632014, India

**Keywords:** cyber–physical system (CPS), Bayesian optimization hyper tuning, quantum convolutional neural network (QCNN), traffic volume prediction

## Abstract

The rapid growth of industry and the economy has contributed to a tremendous increase in traffic in all urban areas. People face the problem of traffic congestion frequently in their day-to-day life. To alleviate congestion and provide traffic guidance and control, several types of research have been carried out in the past to develop suitable computational models for short- and long-term traffic. This study developed an effective multi-dimensional dataset-based model in cyber–physical systems for more accurate traffic-volume prediction. The integration of quantum convolutional neural network and Bayesian optimization (QCNN_BaOpt) constituted the proposed model in this study. Furthermore, optimal tuning of hyperparameters was carried out using Bayesian optimization. The constructed model was evaluated using the US accident dataset records available in Kaggle, which comprise 1.5 million records. The dataset consists of 47 attributes describing spatial and temporal behavior, accidents, and weather characteristics. The efficiency of the proposed model was evaluated by calculating various metrics. The performance of the proposed model was assessed as having an accuracy of 99.3%. Furthermore, the proposed model was compared against the existing state-of-the-art models to demonstrate its superiority.

## 1. Introduction

CPSs [1] have progressively come to include applications [2] and services in communication topologies, actuation devices, computing platforms, and sensing devices. Therefore, advanced techniques, CPS deployment, design, and development have modified this model, opening a path for VCPS—Vehicular Cyber–Physical Systems [3]. However, issues such as energy shortages, environmental pollution, global issues, traffic congestion, and traffic accidents still create a challenging environment. In the last few years, many researchers have tried to solve these; however, construction of the urban traffic infrastructure is extremely slow. For all traffic management, traffic speed prediction plays a significant role with collected data and high-resolution data made available by utilizing an intelligent transportation system. Nevertheless, the availability of high-resolution data is challenging because of traffic-flow stochastic features and the various environmental factors, including rainfall [4].

Traffic prediction based on speed is also an essential process for the ITS—Intelligent Transportation System [5,6]; it produces valuable information for traffic management and vehicle drivers. Furthermore, elevated highways are considered an effective tool of urban transportation that helps expand the road network traffic capacity. The accuracy in speed prediction allows road users on elevated highways to avoid road congestion and guides them to smoother travel. Moreover, most transportation management can optimize traffic resource distribution. Based on the accumulated information from sensors and other sources, researchers have used this information to evaluate traffic data with various techniques such as statistical models, DL—deep learning methods [7], and ML—machine-learning-based approaches [8]. Therefore, among the different statistical techniques, the ARIMA—auto-regressive integrated moving average, and optimized methods based on ARIMA, have been deployed in various research.

Furthermore, government organizations have spent considerable attention and resources on ITSs and the vehicle detector installation process to evaluate and monitor road congestion activities. However, assessing and inspecting congestion is challenging because of different situations, such as data complexity and high volume. Moreover, these problems are compounded due to insufficient analytical systems. Therefore, other visualizations and statistical tools have been utilized for city congestion analysis. The congestion data are collected from CCTVs to monitor for activities such as accidents, theft, over-speeding, etc. Nevertheless, watching more than hundreds of surveillance cameras simultaneously is complex and it leads to a higher chance of missing essential incidents [9].

Traditionally, the execution of various systems with ICT—information and communication technology, advanced measurement in terminals, and remote connectivity have been considered secure because of the proprietary adoption of software and hardware. Furthermore, the advances in ICSs have provided different features and commands to enhance the operational benefits, safety, and reliability that expedite the visualization and interaction of transport companies. Moreover, the ICS network delivers ubiquitous interconnections between the control center and field devices, improving the system’s effectiveness and flexibility in management and monitoring. However, various advancing technologies may enhance the probability of network anomalies and attacks and expose the ICSs to cyber–physical attacks [10].

Smart manufacturing systems have also been used in various applications with CPSs, such as current information and smart grid communication technologies, to communicate, control, and compute to enhance and leverage the monitoring process on physical systems. Furthermore, in Industry 4.0, known as the next-generation revolution of industry, smart manufacturing is already considered significant with actuators and smart sensors being interconnected and associated with industrial systems utilizing advanced techniques that are now widely adopted, reliable, and mature enough with the development of machine devices [11]. Industry 4.0 can be used for various purposes with reliability, good response time, and network latency; moreover, the entire manufacturing process will be executed automatically without the intervention of humans. Therefore, decision making and data transmission will be optimized for obtaining accurate results [12]. Most of the research focuses on traffic prediction based on spatial and temporal factors. Traffic congestion is also caused by other contributing factors such as accidents, weather, location, etc. This research identifies all the other causes of traffic congestion in order to predict with accuracy.

This research mainly focused on analyzing and developing an effective multi-dimensional traffic-volume prediction model in cyber–physical systems. Furthermore, various deep neural network techniques were utilized to enhance the accuracy of the traffic-volume prediction. The QCNN_BaOpt integrates a QCNN and Bayesian optimization model with hyper-tuning for the best parameters to increase the accuracy of traffic flow prediction. The hyperparameter aims to find the parameters of the given model that return the best performance as measured on the validation set, which helps in attaining better prediction for the robust model. Moreover, the model’s accuracy increases as multiple dimensions of the data are considered. The proposed technique was compared with the existing methodology in order to determine the accuracy of the utilized technique. The critical contributions of the current research are as follows:

QCNN_BaOpt for Traffic-Volume Prediction in CPS: The proposed QCNN_BaOpt method is a novel technique that helps to predict traffic volume in cyber–physical systems. In the current research, the QCNN was integrated with Bayesian optimization algorithms to provide an efficient outcome. The other sections in this study are organized as follows: Section 2 summarizes existing techniques to predict traffic volume in cyber–physical systems. Section 3 explains the proposed method for traffic-volume prediction, and the classification report has been elaborated. The results of the proposed techniques and the performance evaluation were compared with other ML techniques and are presented in Section 4. Finally, the conclusions of the current research are summarized in Section 5.

## 2. Literature Review

IBCM-DL is a deep learning technique for the Bayesian model to predict traffic and overcome persistent issues using traditional methods [13]. The new BCM used the IBCM framework. Correlation evaluation was utilized to estimate the applicability between the current and historical traffic flow in the current interval. In the research, GRUNN—gated recurrent unit neural network, RBFNN—radial basis neural function network, and ARIMA—autoregressive integrated moving average were integrated with the IBCM framework. Therefore, real-time traffic-volume data were used in the experiment. However, due to intrinsic flaws, DL—deep learning methods are not always appropriate for traffic prediction.

The STDN—Spatial–Temporal Dynamic Network [14] model was utilized based on spatial and temporal dependency observations. Therefore, a flow-gating mechanism was used to determine the similarity dynamically amidst the various locations through traffic flow. Further, shifted attention method was utilized periodically to manage temporal shifting and long-term dependency. Moreover, region-based traffic identification was extended using a convolutional graph structure. A deep learning-based spatiotemporal neural network model [15] was proposed to predict citywide traffic. A learning structure is devised to analyze the traffic flow patterns considering their topological relationships. The temporal and spatial dependencies were handled using a recurrent neural network and densely connected convolutional network. However, this model failed to deal with other factors that contribute to the road traffic. Nevertheless, sometimes it was incapable of identifying the complicated spatial relationships (dynamic) and nonlinear dependencies (spatial).

The GMAN—graph multi-attention network [16], which concentrated on spatiotemporal factors, was utilized in the study to identify the various traffic-flow scenarios that occurred in different locations with the help of the road-network graph. This technique employs encoder–decoder architecture and multiple spatiotemporal blocks to design the effect of the factors obtained on traffic conditions. Typically, the decoder is used to identify the output, and the encoder receives the input from traffic features. However, long-term predictions still have issues due to error propagation sensitivity and nonlinear temporal and dynamic-spatial correlations.

A neural network-based traffic-forecasting technique was utilized in this study to solve open scientific problems [17]. Therefore, a T-GCN—temporal graph convolutional network was used with the combination of a GRU—gated recurrent unit and GCN—convolutional graph network. Mainly, GCN was utilized to learn complex topological structures for identifying temporal dependence. Further, the T-GCN was applied in traffic forecasting on urban roads. Therefore, the real-time dataset was used in the experimentation and provided an efficient result. Eventually, this model could identify the temporal and spatial features taken from the traffic, and this model can also be employed in other spatiotemporal work. However, the neural network methods used in the sequencing process have limitations because gradient explosion, gradient disappearance defects, and traditional techniques are limited for long-term predictions.

The DNN-BTF—Deep Neural Network-based traffic flow [18] method was utilized to enhance prediction accuracy. The DNN-BTF model uses spatial–temporal features and periodicity characteristics on a daily basis of traffic flow. The traffic flow significance was learned automatically with the attention-based model. Further, the CNN was also utilized in the process with the RNN—recurrent neural network and spatial characteristics. The visualization effects were used to understand and find the traffic flow; however, the variations in the traffic flow make it difficult to analyze in the transportation field—the existing method used in the research was validated with the open-access database PMS for long-term prediction. Nevertheless, most existing techniques have limitations when applying techniques to large datasets and complex environments (nonlinear functions).

HTM—hierarchical temporal memory [19] was used for traffic flow and concentrates on short-term prediction. The procured results were compared with the LSTM—long-short-term memory in online and batch-learning modes with predictive performance. Eventually, the obtained results showed that LSTM and HTM exhibited better results than traditional LSTM, ANNs, and Stacked Autoencoders.

The STANN—spatial and temporal attention technique [20] on traffic flow was used in the research to reveal the spatial dependencies amidst road segments and time steps (temporal dependencies). Therefore, real-time datasets were used in the study for demonstration with the multiple-data resolutions to enhance the prediction precision. Furthermore, existing research was used to understand and identify the spatial–temporal correlation, especially during the traffic network. However, the spatial dependencies of road segments are difficult to determine in the road network due to the distance and smaller neighboring links.

The study used the GCN—graph convolutional network [21] to help extract the spatial and temporal characteristics obtained from the traffic data. Usually, this method is utilized to remove the significant spatial features that help in dynamic feature extraction, compared with the combined GRU and GCN, into one model. Further, an updated strategy was also used for traffic measurements and to procure efficient results. As a result, this technique outperformed other traffic prediction methods. Nevertheless, using the static road network and dynamic traffic measurements has limitations that lead to inaccuracy in traffic flow prediction.

The FDCN—fuzzy DL technique [22] was utilized for identifying traffic flow and fuzzy theory was used with the deep-residual-network model. This reduced the effect of data uncertainty. Further, the fine-tuning and pre-training methods were effectively used to learn FDCN parameters. Therefore, the experimental outcomes showed that the FDCN method was more efficient than other methods.

## 3. Proposed Methodology

The current study mainly focused on traffic-volume prediction and analyzed and developed an effective multi-dimensional data-based traffic-volume-prediction model. The multi-dimensional data gathered from the different sources include location data, weather data, spatial and temporal data, traffic data, and accident data. Exploratory data analysis was performed. Firstly, these data were subjected to the preprocessing stage, where inappropriate information was dropped. Then, the data were encoded, and the KNN Imputation method was used to impute the missing values; feature selection and scaling were performed using the Pearson correlation coefficient and min–max scalar methods, respectively, to speed up the algorithm. The target variables were created using k-means clustering methods. The sampled data were obtained and split for training and testing purposes. The performance prediction used CNN, in which the proposed QCNN_BaOpt was adapted. The proposed QCNN_BaOpt was devised by integrating QCNN and Bayesian optimization. Finally, the trained model was produced using QCNN_BaOpt. Therefore, a novel model architect was created for classifying the traffic level. Figure 1 denotes the prediction method with the convolutional and max-polling layers with the fully connected network.

Q conv and Q pool represent the convolutional layer and polling layer, respectively. Max Pooling is considered a convolution process in which the kernel extracts the maximum value with the CNN—convolutional neural network, taking only important data with the largest data available amplitude-wise. In the proposed prediction method, the classification of sentiments based on new input was considered.

### 3.1. Data Pre-Processing

Pre-processing is essential to transform the data to a more efficient format for processing the information and making predictions easier. Additionally, pre-processing was performed to remove noise and deal with the missing values, enhancing data quality throughout the prediction process. The network-sampling method was utilized to acquire the sample data from large datasets. Exploratory data analysis was used to help discover the patterns, test the hypothesis, and spot unusual activities in initial investigations. The initial pre-processing removed the uncertainties in the data that affect the classification accuracy, such as noise and irrelevant content related to the prediction process, and made the data more efficient and compact for further evaluation. Unused features such as ID, description, number, zip code, and airport code were dropped.

For encoding, the categorical data-encoding method was utilized when the specific levels of categorical encoding features were ordinal. During this scenario, maintaining the order is significant; therefore, the encoding contemplated the sequence of the data. Moreover, in label encoding, every label will usually be converted into values that define integers. Additionally, it is essential to create variables in the operation that include the appropriate traffic flow data. Here, the label encoding method used in the conversion of labels into a numeric form, therefore, helped to convert the data into a machine-readable format. Typically, ML—Machine Learning algorithms can decide an efficient method during the operation of labels. At the same time, it is significant to pre-process the data for structured datasets, especially in supervised learning.

It is crucial to understand the fact that, in various real-world scenarios, datasets can include missing values due to different reasons, and they are continuously encoded as Not a Number (NaN) or zeros, other placeholders, and blanks. Therefore, training a model with a proper dataset is challenging due to the presence of missing values, and, drastically, it impacts the prediction model quality. The primary way to manage this problem is to overcome the observations with missing information or data. Nevertheless, it can risk missing out on data points with meaningful and valuable data. Therefore, it is significant to provide an efficient strategy that helps provide information about the missing values in the dataset. The parameters with the missing values and percentage of null values based on the columns represented include pressure, temperature, humidity, visibility, wind speed, wind chill, and precipitation.

Typically, the k-Nearest Neighbors—kNN [23] method is used to impute the missing data values in the experimentation process. Most of the research conducted by the community and sociologists has explained the safety and security and the attachment of the individual to the neighbors and community. Hence, the relationship with the neighbors is identified via participation in different activities. Similar to this concept, an imputation methodology was used on data with k-Nearest Neighbors—kNN that predicts the neighboring points via distance measurement. The missing values were evaluated using neighboring observations and competing for values. The distance calculation during the existence of the missing values was assessed with the Euclidean distances based on the nearest neighbors as follows:(1)$$d_{\mathrm{xy}}=\sqrt{weight\;*\;squared\;distance\;taken\;from\;the\;current\;coordinates}$$
where,
(2)$$\mathrm{Weight}=\frac{Total\;no.\;of\;coordinates}{No.\;of\;present\;coordinates}$$

The k-NN algorithm’s mathematical formulation is depicted as follows:(3)$$\mathrm{dist}(x,x^{\prime })\;\geq \;\underset{\left(x^{n},\;y^{n}\;\epsilon s_{x}\right)}{\max}dist\;\left(x,\;x^{\prime \prime }\right)$$
where, X—test point, $s_{x}$—k-nearest neighbor of (*x*), and the classifier h() is defined as a function with the label $s_{x}$ as follows:(4)$$h(x)=\mathrm{mode}\;(\{y^{n}:(x^{n},\;y^{n})\;\epsilon s_{x}\})$$
mode (.)—the highest-occurrence selection labels the kNN imputation for missing values.

### 3.2. Feature Selection and Scaling Figures, Tables and Schemes

Eliminating redundant and irrelevant data can deal with issues of dimensionality and identify the features with more relevance to the prediction model. The feature selection and extraction stage include appropriate dataset features for the prediction model with the Pearson Correlation Coefficient (filter method) to perform the tasks [24]. This procedure picks specific and appropriate features and removes irrelevant features from the provided dataset. This filter method is used in the feature-selection process and determines only the relevant feature subset. This model was developed after selecting the appropriate features, and filtering was processed utilizing the correlation matrix, which uses Pearson correlation. Therefore, this specific technique is designed to indicate two similar strong variables. Moreover, this mathematical technique evaluates the relationship between the quantitative variable. The Pearson Correlation coefficient index is expressed as follows:(5)$$r=\frac{\sum \left(x_{i\;\;}-\;\bar{x}\right)\left(y_{i\;\;}-\;\bar{y}\right)}{\sqrt{\sum {\left(x_{i\;\;}-\;\bar{x}\right)}^{2\;\;}\sum {\left(y_{i\;\;}-\;\bar{y}\right)}^{2\;\;}}}$$
where, r—correlation coefficient, $x_{i\;}$—x-variable, $\bar{x}$—x-variable mean values, $y_{i\;}$—y-variable, $\bar{y}$—y-variable mean values.

Figure 2 represents the correlation of independent variables with the output variable MEDV as stated in Equation (5). Features which had a correlation of above 0.9 (absolute value) with the output variable were considered for prediction. Target variables which were highly correlated with other variables were only accepted. The heatmap helps to visualize the strength of relationships between the numerical variables. The labels on the *x* and *y* axes are the parameters of the multi-dimensional data such as location data, weather data, spatial–temporal data, traffic data, and accidental data.

#### 3.2.1. Target Variable

The k-means clustering technique is the simplest and most popular unsupervised ML algorithm used to create the target variables. These algorithms generate inferences from the obtained datasets utilizing only input vectors in the absence of labeled outcomes, known as referring. In other words, the k-means algorithm predicts the k-number of centroids to allocate the nearest cluster data point during the centroid presence. Additionally, the target values were created using the abovementioned Python code in the process. Further, the graphical representation of the label creation is depicted in Figure 3 with the number of classes.

#### 3.2.2. Feature Scaling

The standardization of feature scaling occurs in data pre-processing, is employed for independent variables, and helps in data normalization in a particular range. Therefore, it helps to speed up the mathematical formulations in an algorithm.

Standard Scaler—the standard scaler transforms the acquired information or data if they have a standard deviation of 1 and a mean value of 0 that standardizes the provided data. Therefore, standardization is considered useful for data with negative values and organizes the data or information in a standard normal distribution. Further, these follow SN—Standard Normal Distribution with the mean value of 0 and help scale the data to unit variance. The mathematical formulation of the standard scaler is expressed as follows:(6)$$X_{\mathrm{new}}=\frac{x\;-\;\mu }{\sigma }$$
where, μ—mean value, σ—standard deviation.

Minmax Scaler—The minmax scaler in the experiment measures the data features presented in the range of [0, 1] or [−1, 1] in case of negative values in the provided dataset. Therefore, this scaling process concentrates on the inliers in the range of [0, 0.005]. During the outlier’s existence, the standard scaler will not ensure the standard deviation and empirical mean values, leading to shrinking feature values in the specific range. In the current experiment, the robust scaler was used to eradicate the outliers, and either the minmax scaler or standard scaler was used for dataset preprocessing. The mathematical formulation of the minmax Scaler is denoted below:(7)$$X_{\mathrm{new}}\;=\;\frac{x\;-\;x_{min}}{x_{max\;-}x_{min}}$$
where, $x_{min}$—minimum values, $x_{max}$—maximum values.

The imbalanced data were checked and managed with the oversampling method to treat the imbalance problems. Therefore, the imbalanced data denote the dataset types with the target class, including uneven distribution, especially in one class label with a significant number of observations and others with low observation numbers.

### 3.3. Traffic-Volume Prediction Using QCNN_BaOpt

Here, the QCNN-based prediction model is represented by the high-feature dataset for traffic-volume prediction. Multi-dimensional datasets with location data, weather data, spatial and temporal data, traffic data, and accidental data were considered for traffic prediction. The QCNN was integrated with Bayesian optimization with hyper-tuning parameters for the proposed strategy, which inherits the CNN and QCNN’s structural design.

#### 3.3.1. Architecture of the Proposed System

The QCNN_BaOpt model was proposed model in this study. It integrates a quantum convolutional neural network and Bayesian optimization model with hyper tuning for the best parameters to increase the traffic flow accuracy. The main focus of these hyperparameter optimizations in ML is to analyze and find the hyperparameters for a provided ML algorithm that acquires the best performance, which is evaluated on a validation set. Each iteration was executed with the target values. The Bayesian optimization and the utility function help to measure the accuracy of the prediction method. The architecture of the proposed methodology, QCNN_BaOpt, is illustrated in Figure 4.

The convolution layer (Q conv—convolutional layer) is considered the first layer in a convolutional neural network (CNN). The matrix input is provided with the dimensions (height (h1) * weight (w1) * diameter (d1)). The kernel matrix is represented with the dimensions (height (h2) * weight (w2) * diameter (d2)) that form the 3D matrix and finally the output layer is represented with the dimensions (height (h3) * weight (w3) * diameter (d3)). Every represented kernel position was multiplied by the input matrix and summed up with the value that defines the position in the specific output matrix.

Generally, in image processing, CNN contains different layers, known as interleaved layers. For example, a feature map consists of 2D or 3D array pixels with an intermediate presented in every layer. The pixel values are computed by convolution layers represented as derived by a linear combination with the preceding map equation as follows:(8)$$z_{k,j}^{l}=\overset{wi}{\underset{c,d=1}{\sum }}wi_{c,d=1}\;z_{k+c,\;j+d}^{\left(l\;-\;1\right)}$$

Here, the weights are denoted as $wi_{c,d\;}$ and form a wi * wi matrix.

The pooling layer minimizes the size of the feature map by considering the maximum value obtained from a contiguous pixel that is followed by an activation (nonlinear) function. After the size of the feature map turns sufficiently small, the obtained outcome is estimated from a certain function that relies on the rest of the pixels, which is fully connected. Therefore, the function that is fully connected and which provides weights is eventually optimized by large, trained datasets. On the contrary, variables include the pooling and convolution layers with weight wi matrices known as hyper-parameters that remain the same for a particular CNN. The critical characteristics of CNNs are therefore translationally pooling and invariant convolution layers. Hence, every layer is distinguished by certain constant parameters representing system size, while sequential data-size minimization represents a hierarchical structure.

Pooling layer (Q pool)—the pooling concept consists of two types: max pooling and average pooling. The main use of the pooling layer is to minimize the number of parameters in the input tensor and help minimize overfitting issues in the proposed technique. Further, extracting the specific features taken from the input tensor minimizes the computation efficiency of the technique. The max pooling and the matrix were estimated with the kernel size (*n* * *n*). Each position was represented as a max value and denoted in the output matrix-specific position. On the other hand, the average pooling with the kernel size (*n* * *n*) was transformed into the matrix. Every position was denoted as average values and specified in the output matrix specific position. The output of the conventional layers was evaluated using Equation (9).
(9)$$\mathrm{Conv}\_\mathrm{Layer}\_\mathrm{Output}\;=\frac{\mathrm{input}\;\left(i/p\right)-\;\mathrm{kernel}\_\mathrm{size}\;\left(\mathrm{kl}\right)+2\;*\;\mathrm{padding}\;\left(\mathrm{padd}\right)}{\mathrm{stride}\;\left(\mathrm{str}\right)}+1$$

A max pooling layer accompanies the techniques utilized in the study using the pooling layer. These should be same as the convolutional layer and the max-pooling layer in the kernel size, which is denoted as 2, 2 with stride 2 and the output was calculated using Equation (9). The second convolutional layer was identified in a similar way.

Fully Connected Layer (FC)—here, the fully connected layers were simplified from the previous layers in the network. The provided input was connected fully with the layers in the current research. The output procured from the flattened pooling layer was given to the fully connected layer. The parameters in the fully connected layer were measured as follows with a training example. The provided input (i/p) is as x11, x22, and x33 that are 3D with prediction issues and inputs based on the scenario. Firstly, the hidden layers are represented with certain units and calculated with the sigmoid function, which is the activation function as in Equation (10).
(10)Activation_function = σ(Wdz + a)

Here, Wdz represents the dimension with the parameters represented as a; the total first hidden layer parameters are represented and the values are calculated based on it. The dimension of Wdz is represented in Equation (11).
(11)$$Wd^{l}=\left[u^{\left[l\right]},\;u^{\left[l-1\right]}\right]$$
(12)$$a^{\left[l\right]}=\left[u^{\left[l\right]},\;1\right]$$

*l*—defines the layer; $u^{\left[l\right]}$—defines the l-layer units and the parameters of the layers are represented using Equations (12) and (13).
(13)$$\mathrm{Parameters}\_p=u^{\left[l\right]}*\;u^{\left[l-1\right]\;}+u^{\left[l\right]}$$

#### 3.3.2. Model Creation and Training

Here, the model creation was carried out, the prediction model was availed, and the result was formulated using the selected feature dataset. The data were split into two forms: one for training and the other for testing, wherein the training data comprise 70% and the testing data comprise 30% of the pre-processed data. After splitting, the model used the random forest technique code.

Integration of quantum convolutional neural networks with Bayesian optimization Hyperparameter Tuning—the QCNN [25]—quantum neural networks use hierarchical structures such as tree-like along with the qubits number taken from a preceding layer, which is minimized by two factors for the subsequent layer. Normally, these architectures include O(log(n)) layers represented for input qubits (n). Further, the minimization of the qubits number is analogous and denotes the pooling operation presented in convolutional neural network (CNN). The QCNN_BaOpt framework has distinct features and acts as the variance translational element that drives the parameterized quantum gate blocks to be similar in a given layer. The QCNN quantum state layer results are represented using Equation (14).
(14)$$|\psi i\;\left(\theta i\right)\mathrm{ih}\;\psi i\;\left(\theta i\right)|=\mathrm{TrBi}\;(\mathrm{Ui}\;(\theta i)\;|\psi i-1\mathrm{ih}\;\psi i-1\;|\;\mathrm{Ui}\;(\theta i{)}^{ϯ})$$

TrBi denotes the partial trace operation in subsystem and U_i_ denotes the parameterized unitary gate-operation that consists of the pooling part and the convolution gate as denoted in Equations (15) and (16) respectively.
(15)$$\mathrm{Bi}\;\in \;C\;\frac{n}{2i}$$
(16)|ψ0i|=|0i ⊗ n|

The parametrized quantum circuit structure is referred to as ansatz. The architecture QCNN_BaOpt and two-qubit quantum circuit blocks (*U_i_*) were utilized with similar pooling and convolution layers. In this study, two convolutional layers were used with the two-qubit gate and certain parameters that were represented as a layer include l_i_ > 0; here, the convolution filter is independent and 1-pooling operation was used to increase the number of parameters utilized with the Bayesian optimization. Hyperparameter tuning is considered a challenging issue in machine learning and needs expert skills that concentrate on the deep learning model and include different parameters. Recently, various studies have been conducted to execute the process without human intervention with different automated search techniques. In the current research, the hyperparameter was taken utilizing Bayesian optimization due to the increased success rates. Here, θ1, θ2, θ3, ………θn are defined as the hyperparameters and the optimization problem is denoted by Equation (17).
(17)$$\theta^{*}={\;\mathrm{argmin}}_{\theta \in \Phi \;}f\left(\theta \right)$$f(θ)—Objective score, Φ = Φ1 × Φ2 × Φ3 ×………Φm—hyperparameter search space.

Bayesian optimization has the capability to run different models with various hyperparameter values; however, it analyzes the information from the previous model to choose the hyperparameter values to develop the newer model. The gathered evaluation results were used to develop a probabilistic model mapping hyperparameters to a score probability that focused on the objective function represented in Equation (18).
(18)P(Score (z) |Hyperparameters(y))

Moreover, this model is known as a surrogate mainly for the objective function and is denoted as p(z|y). Here, the surrogate is considered simple to optimize when compared with the objective function that has been represented below. Further, the Bayesian technique was carried out by identifying the hyperparameters to estimate the actual objective function by choosing hyperparameters that provide better performance on the surrogate function.

In the proposed hybrid method, Bayesian optimization plays an important role and this is considered an effective technique in the research due to the global optimization form of black box functions that are expensive to evaluate. Therefore, it depends on the distribution of querying over certain functions denoted by a surrogate method that is relatively cheap. Due to the limitations that persist in the existing research on Gaussian processes, Bayesian optimization was combined with the QCNN model in the research to study adaptive basis functions in order to procure an effective outcome. The Bayesian optimization Algorithm 1 is stated below.
**Algorithm 1** BO—AlgorithmStep 1: For u = 1, 2, 3, 4 ….Step 2: The acquisition function is optimized and represented as v over fu function      to find yt.           yt=arg maxv (y | S1:t−1)Step 3: Objective function sampling with the equation—zt=fu (yt).Step 4: Data augmentation—S1:t={S1:t−1 (yt, zt)}Step 5: Update the posterior function fuStep 6: Quantum data encoding function.

Quantum Data Encoding—various machine learning techniques were utilized in the study for input data transformation, denoted as *X* into various space. This is also known as a feature map. A similar analogy can in the employed in quantum computing for quantum feature mapping. The transformation function and the derived quantum feature map are denoted using Equations (19) and (20) respectively.
(19)$$\varphi \;:\;X\;->\;X^{\prime }$$
(20)φ : X −>H
where *H* denotes the Hilbert space. Feature mapping plays an important role in ML on classical data. The quantum feature map is stated in Equation (21).
(21)x ∈ X−>|φ(x)i| ∈ H

The equivalent unitary transformation function *U*_φ_(x) in the initial state |0_i_ ⊗ n| is employed further in Equation (22). The different structures are presented based on *U*_φ_(x) for classical input encoding (x) in the quantum state.
(22)Uφ(x) | 0i ⊗ n =|φ(x)i|
where n represents the qubit numbers.

Quantum Layer creation with connections—in this experiment, the quantum structure was created with the classical layers with the SoftMax activation function, which defines a quantum circuit along with two qubits. Therefore, certain parameters on layers, including qnode, weight of the circuits, and output functions, were specified in the layers with the specific range of N_QUBITS as defined in this study.

Connecting convolution with quantum layers—the CNN model was evaluated with the provided input layer, quantum layer, and output layer with the optimizer learning rate of 0.01. Moreover, the accuracy of the utilized QCNN technique was estimated with the validation loss, loss, accuracy, and validation accuracy.

## 4. Results and Discussion

The QCNN_BaOpt model was evaluated with the performance evaluation parameters with MAE—mean Absolute Error and RMSE—root-mean-square deviation comparison. Eventually, the proposed MAE was procured with better results compared with the other existing MAE techniques.

### 4.1. Experimental Setup

The implementation of the proposed method was experimented in Python using the jupyter notebook with a PC running Windows 11 OS, 4 GB RAM, and with an Intel i5 core processor.

### 4.2. Dataset Description

The current research used different datasets to predict the traffic flow accuracy with the proposed design. Here, the traffic count dataset that gathered traffic count for every junction was used with certain dimensions and parameters. The dataset dimensions such as location data, weather data, spatial and temporal data, traffic data, and road accident data were acquired and merged to experiment with the proposed methodology. The location data parameters included the latitude and longitude GPS coordinates of the start and end point features. The weather data included temperature in Fahrenheit, wind chill in Fahrenheit, humidity in percentage, pressure in inches, visibility in miles, wind direction, wind speed in mph, precipitation in inches, and weather conditions specified as drizzle, rain, light rain, overcast, clear, partly cloudy, mostly cloudy, scattered clouds, snow, light snow, and/or haze. The spatial–temporal data included time stamps such as start time and end time (specifies the time of the accident) and latitude and longitude information. The traffic data included junction, stop, traffic signals, and traffic calming. Finally, accidental data included crossing, severity, which showed the impact of accident on traffic ranging from 1 to 4, civil twilight, nautical twilight, and astronomical twilight. Here, the datasets were taken from https://www.kaggle.com/datasets (accessed on 27 November 2022) and accident datasets were considered for the proposed systems’ experimentation process.

### 4.3. Performance Measures

The MAE and RMSE metrics were used to evaluate the performance of the proposed method. Figure 5 represents the graphical representation of the MAE and RMSE comparison as illustrated for the existing system and proposed system.

MAE: The MAE is the mean absolute error, which is the difference between the estimated and observed values.
(23)$$\mathrm{MAE}-\frac{\sum_{i=1}^{n}\left|\;y_{i}-x_{i}\right|}{n}$$
where, $y_{i}$—prediction, $x_{i}$—true value, and *n* is total number of data points.

RMSE: RMSE is the root mean square error, which terms the difference between the square root of estimated and observed values.
(24)$$\mathrm{RMSE}-\sqrt{\frac{\sum_{i=1}^{N}{\left(x_{i}-t\;\right)}^{2}}{N}}$$
where, *i*—variable, *N*—non-missing data points, $x_{i}$—observation time series, and *t*—estimated time-series.

### 4.4. Performance Analysis

The performance comparison of the proposed QCNN_BaOpt with CNN was processed and illustrated. The training and validation accuracy and loss, classification report, and the accuracy were considered for the performance evaluation of the existing and proposed systems.

#### 4.4.1. Training and Validation–Accuracy and Loss

In the training and validation accuracy, the training accuracy is typically the most significant subset created out of the multi-dimensional dataset that is used to fit the model, whereas the validation accuracy is later used to evaluate the model to perform the model selection. The training and validation accuracy are depicted in Figure 6 for existing—CNN and proposed—QCNN_BaOpt respectively.

The training loss indicated how well the model fit the training data, while the validation loss suggested how well the model fit the multi-dimensional dataset in the proposed system. The graphical representation of the training and validation loss is depicted in Figure 7 for existing—CNN and proposed—QCNN_BaOpt respectively.

#### 4.4.2. Classification Report

The existing and the proposed model procured 0.78 and 0.99 accuracy, respectively, and the classification report is provided in the Table 1 and Table 2 below with certain performance evaluation metrics, including precision, recall, F1 score, and support.

Compared with the existing and proposed methods for training and validation accuracy and loss estimation, the proposed method had a better outcome with less validation loss. Eventually, the existing models were compared with the proposed model in order to estimate the accuracy. Further, the graphical representation of the proposed QCNN_BaOpt procured better results compared with the other existing models such as CNN and RF, as depicted in Figure 8.

## 5. Conclusions

The development of urbanization, social economy, and road traffic have extended, and quantity-possessing vehicles have started to increase, which has resulted in rapid growth, especially in urban traffic. The paper mainly focused on a novel method for traffic-volume prediction using QCNN_BaOpt, where the integration of a quantum convolutional neural network and Bayesian optimization with hyper-tuning parameters was used. Further, we analyzed and developed an effective multi-dimensional data-based traffic-volume prediction model in cyber–physical systems to improve the accuracy. The proposed QCNN_BaOpt model was compared with the existing machine learning methods and CNN and procured 0.993 accuracy results. The classification report of the proposed method was evaluated with certain performance evaluation metrics. The accuracy of the model prompts for was measured for the prediction of traffic in high-traffic-density and real-time environments. The prediction model could be incorporated with adaptive learning to find the patterns in the missing traffic data, which provides references for future applications such as weather predictions.

## Figures and Tables

**Figure 1 sensors-23-01485-f001:**
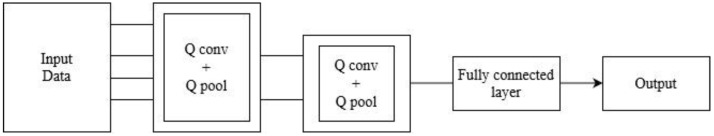
Prediction method with Q conv, Q pool, and full connection layer.

**Figure 2 sensors-23-01485-f002:**
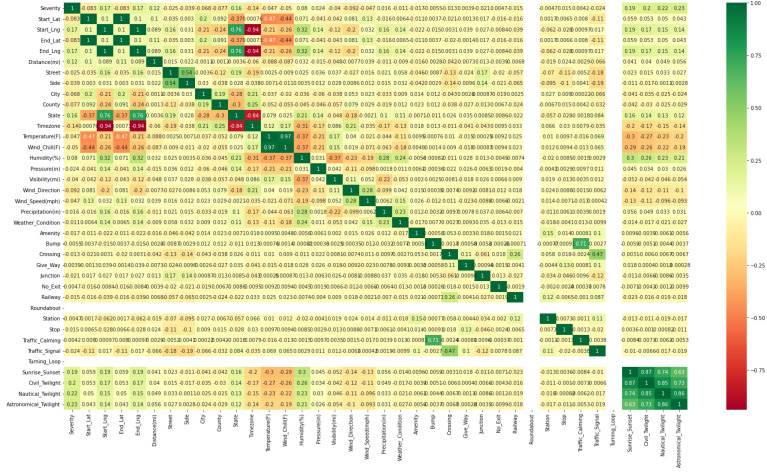
Correlation heatmap with independent and MEDV output variables.

**Figure 3 sensors-23-01485-f003:**
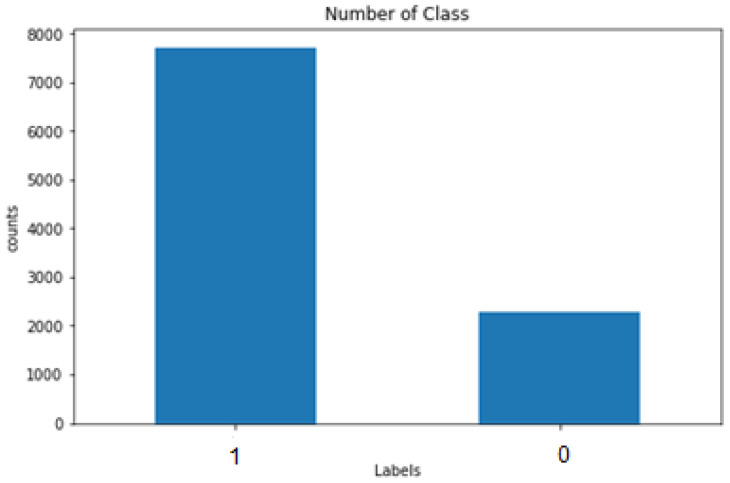
Graphical representation of the creation of the target variable.

**Figure 4 sensors-23-01485-f004:**
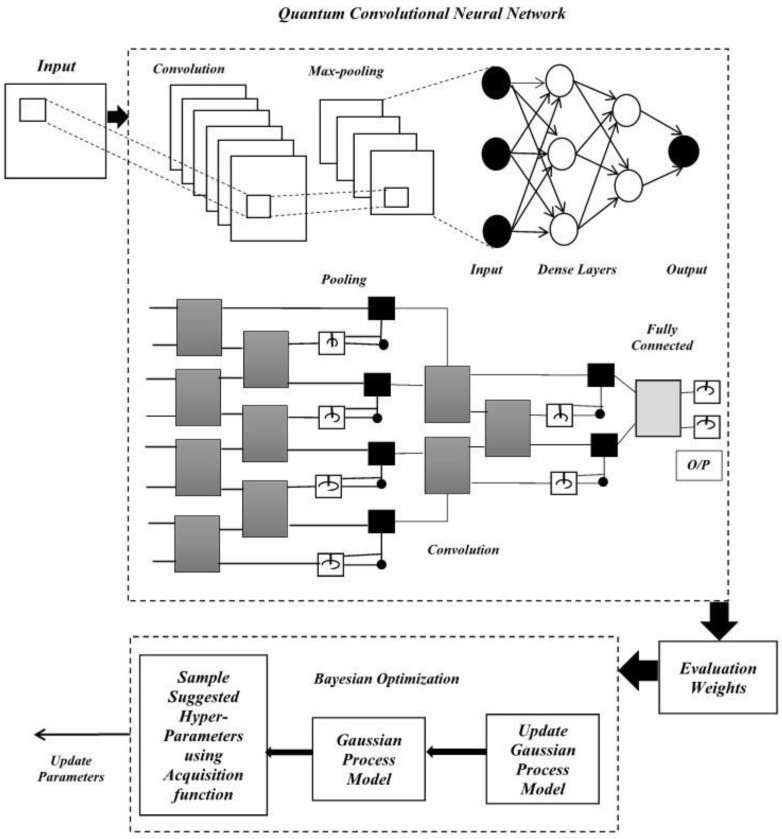
Architecture of the proposed methodology.

**Figure 5 sensors-23-01485-f005:**
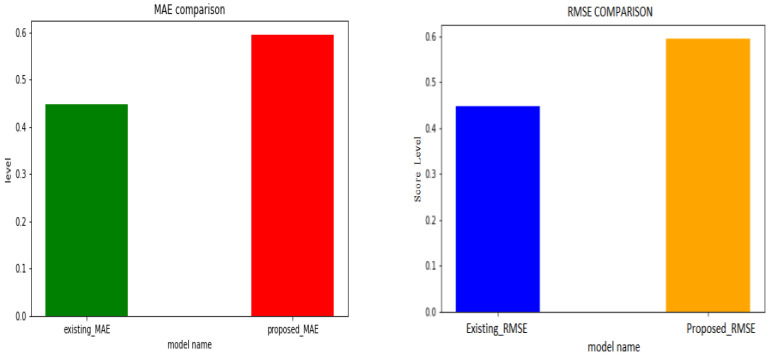
Performance evaluation of MAE and RMSE.

**Figure 6 sensors-23-01485-f006:**
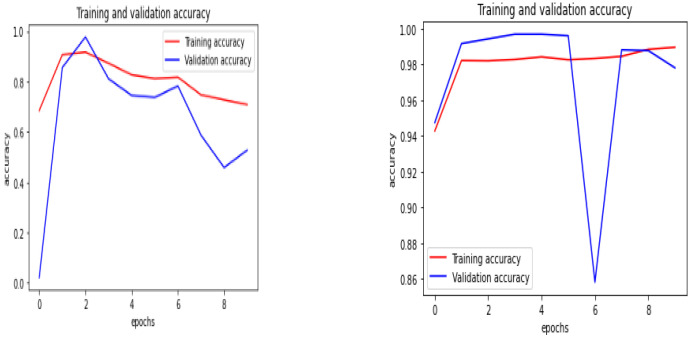
Training and validation accuracy of the existing and proposed systems.

**Figure 7 sensors-23-01485-f007:**
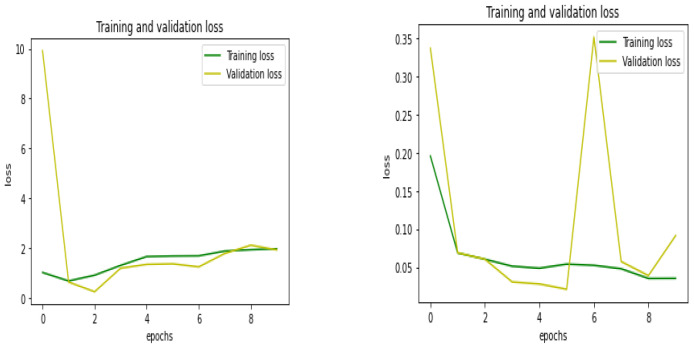
Training and validation loss for the existing and proposed systems.

**Figure 8 sensors-23-01485-f008:**
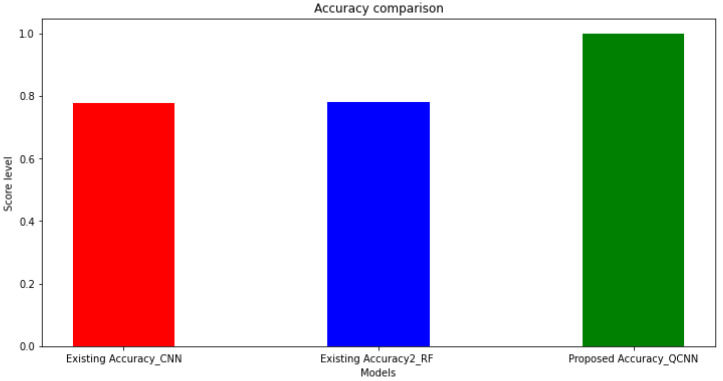
Accuracy comparison for the proposed method.

**Table 1 sensors-23-01485-t001:** Classification report for the existing system (CNN).

	Precision	Recall	F1-Score	Support
0	0.50	0.99	0.66	657
1	1.00	0.72	0.83	2343
5	0.00	0.00	0.00	0
Accuracy	-	-	0.78	3000
Macro_avg	0.30	0.34	0.30	3000
Weighted_avg	0.89	0.78	0.80	3000

**Table 2 sensors-23-01485-t002:** Classification report for the proposed system (QCNN_BaOpt).

	Precision	Recall	F1-Score	Support
0	1.00	0.99	0.98	657
1	0.99	1.00	1.00	2343
Accuracy	-	-	0.99	3000
Macro_avg	0.99	0.99	0.99	3000
Weighted_avg	0.99	0.99	0.59	3000

## Data Availability

Not applicable.

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
