# Peer review of "QCNN_BaOpt: Multi-Dimensional Data-Based Traffic-Volume Prediction in Cyber–Physical Systems"

_sensors, 2023, doi:10.3390/s23031485_

Round 1

Reviewer 1 Report

The paper proposes a hybrid model consisting of Quantum Convolutional Neural Network and Bayesian optimization for traffic flow prediction. The performance of the model is high. However, the paper needs major revision as follows:

1- The abstract statement are two general and the problem should be highlighted better, why the proposed method is needed?

2- English needs to be revised 

3- the problem in the introduction section is scattered and not focused. There should be a paragraph shows the gap 

4- there should be a paragraph states the contribution clearly and the importance of the contribution at the end of the introduction section

5- The related work can be enhanced by adding research talking about spatiotemporal traffic flow model and models handling high traffic variance and overdispersion -- Check IEEE Transactions on Intelligent Transportation Systems

6- In the data section, put a graph to show the pattern in the data

7- the labels on x and y axes in figure 2 are not clear and figure 2 should be explained, what is heatmap and all labels in the figure 

8-how does the proposed model deal with the high variance of traffic speed or volume ?

9- the figures resolution should be improved 

10 - explain figure 6 and figure 7, and 8

11- highlight future directions in research 

12- what are the sources of uncertainity of the data used in this research? show how the FDCN - fuzzy DL reduce the uncertainity of the data used in this research?

Author Response

Thank you for the valuable comments. We found them really helpful and tried to answer all the comments to the best of our ability.

Reviewer 2 Report

Glad to review the paper (sensors-2094344). A Quantum Convolutional Neural Network-based traffic volume prediction model is proposed. Bayesian approach is integrated into the prediction model to refine the parameters. Data samples are enough to convince. All in all, the paper is interesting. However, the language is not good as per the academic level. There are some comments to help the authors to improve further.

Major problems:

(1)   I did not find out the relationship between sensors and precision models. Are the prediction data derived from sensor data collection? Whatever, it was never introduced at all.

(2)   Why does the prediction execute in the Cyber-Physical System? Otherwise, what about the other scenarios?

(3)   Re L82-87, I did not understand the motivation of this study. Why the proposed method (the multi-dimensional traffic volume prediction model) is feasible to handle the prediction precision?

(4)   Considering the references [7] and [8] on L47, why did these approach lose the ability of accurately predicting? In fact, Deep Learning [7], and Machine learning are state-of-the-art techniques currently. As a doubt for me, the contributions are unexpectedly for the current precision development tendency. The authors should have explained clearer.

(5)   P86, I did not think using A Quantum Convolutional Neural Network is a key contribution. It is only presented to apply an existing method for prediction case, in my view, which is far away from the so-called real contribution. The right way is that the authors should find out the advantages of the proposed model compared with the previous works.

(6)   Section 2 (Literature Review) is available but what’s the effectiveness on carrying out the current work? The authors should summarize the deficiencies of them rather than only describing. They should be a research fundamental to inspire the newly-proposed approach. However, it ended in such rush.

(7)   L174-175, how did the data, such as “location data, weather data, spatial and temporal data, traffic data and accident data”, use in the prediction model?

(8)   My doubt is how to obtain the results 0.777 and 0.993 on L566. In fact, the outcomes are not corresponding to Tables 1 and 2. At least, the significant digits after the decimal point should revise.

(9)   I am not aware of why the big difference exists between 0.777 and 0.993 for “recall” and “Weighted_avg”. The authors should explain.

Minor problems:

(1)   P185, the full stop behind “Figure 1” should be removed. The similar errors need be corrected. Figures 2-7 with the full stops are all wrong in the whole paper. Besides, “the” before “Figures 7” should be eliminated, too.

(2)   Figure 1 is too dim. Figure 2 is dim.

Author Response

(The authors gave the same response as above.)

Round 2

Reviewer 1 Report

What is the difference between the proposed method and The methods in 

Negative binomial additive models for short term traffic flow forecasting in urban areas.

And the paper 

Space time multivariate Negative binomial regression for urban short term traffic volume prediction 

Author Response

We appreciate the valuable comments from the reviewer. We tried our best to answer the comments.

Reviewer 2 Report

Dear Authors:

Thanks for your revised paper. Most of comments are responded in a reasonable manner. It is good. However, the 1st question was missed partly. That is, "I did not find out the relationship between sensors and precision models." The authors should explain why the study is within the scope of this journal. Thus, the authors still need to make more revisions besides the language issues. 

Author Response

(The authors gave the same response as above.)
